# Effects of Cherry (*Prunus cerasus* L.) Powder Addition on the Physicochemical Properties and Oxidation Stability of Jiangsu-Type Sausage during Refrigerated Storage

**DOI:** 10.3390/foods11223590

**Published:** 2022-11-11

**Authors:** Qingquan Fu, Shangxin Song, Tianlan Xia, Rongrong Wang

**Affiliations:** School of Food Science, Nanjing Xiaozhuang University, Nanjing 211171, China

**Keywords:** cherry powder, sausage, physiochemical property, oxidation stability, volatile flavor substance

## Abstract

Effects of different levels (1%, 3% and 5%) of cherry powder on the physiochemical properties and antioxidant activity of Jiangsu-type sausages were investigated at 4 °C for 30 days. The results show that the sensory evaluation values and physicochemical properties of the sausages had no significant differences compared to the control group when cherry powder addition was 1%, and the alcohols, aldehydes and esters were increased after the addition of cherry powder improved the flavor of sausages. However, higher concentration of cherry powder (3% and 5%) exerted adverse influences on sensory evaluation values and physicochemical properties of sausages compared with the control. The addition of cherry powder could better inhibit lipid and protein oxidation of sausages, and the cherry powder concentration has a positive correlation with its effect on the inhibition of lipid and protein oxidation. In addition, cherry powder could effectively control TVB-N values of sausages during chilled storage. All these results indicate that 1% cherry powder could not only guarantee the physicochemical properties of sausages, but also inhibited the oxidation of sausages during chilled storage.

## 1. Introduction

As a popular Chinese meat product in Jiangsu Province, Jiangsu-style sausage is favored by both domestic and foreign consumers because of its unique sensory characteristics and rich nutritional value. However, because the fat content of sausage products is generally very high, lipid oxidation is prone to occur during processing and storage. Rancidity and peculiar odors caused by excessive fat oxidation seriously affect the quality and shelf life of sausages [1]. Lipid oxidation also causes protein oxidation in sausages, which significantly affects the texture, biological function, nutritional value, and other eating qualities of sausages [2]. Therefore, the addition of antioxidants can inhibit oxidation and maintain the quality of sausages [3]. 

In recent years, synthetic antioxidants such as butylated hydroxyanisole (BHA), butylated hydroxytoluene (BHT), and tert-butylhydroquinone (TBHQ) have been widely used in sausage production to inhibit the oxidation of meat products. This is because of their low cost and high efficiency [4]. However, an increasing number of studies have shown that the long-term use of synthetic antioxidants can induce gastrointestinal discomfort, respiratory disorders, skin problems, and adverse reactions in the nervous system. In serious cases, it can even increase the risk of cancer and malformation [5,6]. Therefore, synthetic antioxidants pose a potential safety risk to human health. Compared with synthetic antioxidants, natural antioxidants are becoming increasingly preferred by researchers and manufacturing enterprises. Currently, there is an urgent need to develop safe, efficient, and environmentally friendly antioxidants that are natural. 

Cherry (*Prunus cerasus* L.), also known as peach, is a fruit trees of genus Lycaceae in the Rosaceae family [4]. The ripe fruit is bright red and sweet, making it one of the most popular high-end fruits among consumers. Cherry powder is produced from cherry juice via spray-drying. Some studies have shown that cherry juice is rich in polyphenols, such as anthocyanins, gallic acid, and quercetin [7]. The monomeric anthocyanin, total polyphenolic content, and antioxidant capacity of cherry juice were within the ranges 350.0–633.5 mg/L, 1510–2550 mg/L, and 20.0–37.9 mmol/L [8]. Therefore, it has a high antioxidant capacity, as well as anti-inflammatory, antimicrobial, anti-platelet aggregation, anti-tumor activity, and antiapoptosis [9]. Furthermore, it can inhibit tumor cell growth, and reduce the incidence of chronic diseases and cancer [10,11]. Currently, studies on cherries have mainly focused on the extraction of polyphenols and on antioxidant activities. For example, Nowak et al. [12] used varied methods to extract polyphenols, such as flavonoids and anthocyanins, from cherries. Zhang et al. [13] compared the antioxidant properties of different cherry varieties following juice preparation. However, until now, there have been no reports on the addition of cherry juice or cherry powder to meat products, particularly sausage products. Therefore, in this study, the physical and chemical properties of Jiangsu-type sausage including pH, water content, color, and total volatile base nitrogen, as well as oxidation stability of lipid oxidation and protein oxidation, were measured by adding varying concentrations of cherry powder. In addition, the color, texture, and volatile flavor substances were assayed to evaluate the influence of cherry powder on the eating qualities of sausage. The findings will help to promote the use of natural antioxidants in the sausage industry. 

## 2. Materials and Methods

### 2.1. Materials and Chemicals

Cherry powder was purchased from Tianjin Xiugu Biotechnology Development Co., Ltd. (Tianjin, China). Hind leg pork, fat, salt, white sugar, monosodium glutamate (MSG), glucose, isovitamin C sodium, sodium nitrite, and Chinese liquor were provided by Nanjing Yurun Food Co., Ltd. (Nanjing, China). Magnesium oxide, sodium dodecyl sulfate, and glacial acetic acid were analytically pure, and purchased from Nanjing Chemical Reagent Co., Ltd. (Nanjing, China). Dithiobarbituric acid (purity > 98.5%) and 1, 1, 3, 3-tetraoxyethyl propane (purity > 99%) were purchased from Shanghai Chemical Reagent Co., Ltd. (Shanghai, China). Cyclohexanone was chromatographic grade and purchased from Aladdin Reagent Company (Shanghai, China).

### 2.2. Preparation of Jiangsu-Style Sausage

The recipe for Jiangsu-style sausage was as follows: 170 kg hind leg pork, 30 kg backfat, 6.8 kg salt, 13.7 kg sugar, 0.3 kg MSG, 1.2 kg Chinese liquor, 0.4 kg glucose, 0.4 kg isovitamin C sodium, 0.006 kg sodium nitrite, and 2 kg ice water. The pork meat was ground using a JR200 grinder with an 8 mm perforated plate (Hebei Xiaojin Merchinery Manufacturing Inc. Plant, Shijiazhuang, China), and the backfat was cut into cubes of 6 × 6 × 6 mm^3^. The sausages used in the experiment were divided into four groups. A version without the addition of cherry powder was used as the control group, and sausages with additions of 1%, 3%, and 5% cherry powder were used as the experimental groups. The lean meat, all auxiliary ingredients, and ice water were blended in a blender and stirred for 5 min using a JB650 vacuum mixer (Hebei Xiaojin Merchinery Manufacturing Inc. Plant, Shijiazhuang, China). After mixing evenly, the fat was added and stirred for another 20 min. The 26 mm collagen casings were filled using a VF612 vacuum filler (Handtmann Group, Bolin, Germany), and the lengths of the sausages were 10 cm. Finally, the sausages were placed in a smoke oven (Fessmann Group, Bolin, Germany), and dried at 48 °C for 4 h, then at 52 °C for 30 h. After cooling completely, the sausages were vacuum packaged and stored in a refrigerator (Haier Group, Qingdao, China) at 4 °C for 30 days. The samples at 0, 10, 20, and 30 days were used to determine the relevant indicators.

### 2.3. Evaluation of Physicochemical Properties 

#### 2.3.1. The Determination of pH Value and Water Content

Ten grams of the pre-broken sausage samples was added to 100 mL of 0.1 mol/L potassium chloride solution. Thereafter, the mixtures were homogenized using an IKA 25 homogenizer (Ultra Turrax, Staufen, Germany) at 2000 rpm for 1 min, and pH was measured using an XPR106DUHQ pH meter (Mettler Toledo, Schwerzenbach, Switzerland). The average value of the final result for five runs was obtained in parallel measurements. The moisture content was determined using a near-infrared fast-measuring instrument (FoodScan, Hillerod, Denmark) used in determining food composition. The tray was rotated once for each measurement, and the final result was the average of the three measurement results. 

#### 2.3.2. Instrumental Color Measurement

The *L** value (brightness), *a** value (redness), and *b** value (yellowness) of the color on the sausage surface were determined using a colorimeter (CR 300, Minolta, Osaka, Japan). The sausages were smashed with a crusher, and the samples taken were evenly mixed and then spread flat on Petri dishes. The colorimeter was used to randomly measure at eight points in different positions with the illuminant D_65_. Three sausages were taken from each treatment group for measurement, and the average value of the measurement results for all samples was calculated. A white porcelain plate was used to calibrate the chromometer prior to sample determination.

### 2.4. Measurement of the Texture Profile

The samples were cut into 20-mm high cylinders, and the casings were removed. Hardness, springiness, cohesiveness, adhesiveness, and chewiness were measured using a TA-XT2i texture analyzer (Stable Micro System, Surrey, UK). The average value was obtained for each sample from eight rounds in parallel. The texture analysis conditions were as follows—determination mode: TPA; probe type: P50; trigger force: 1000 N, before the determination of speed: 2 mm/s; determination of speed: 1 mm/s, after the determination of speed: 2 mm/s; pressure distance: 30%; two intervals: 5 s; determination of temperature: 25 °C.

### 2.5. Sensory Evaluation

Sensory evaluation of the sausages was performed according to the methods of Yi et al. [14] with suitable modifications; five men and five women who are students, teachers, and members of an organoleptic evaluation group of a company’s technical personnel were invited. The sausage samples were placed in a cooking boiler and steamed for 20 min. The samples were removed and cooled to room temperature. Finally, they were sliced into 30-mm thick pieces for evaluation. In total, there were 9 points of sensory evaluation, and a comprehensive evaluation was carried out according to color, flavor, taste, and overall acceptability. The samples were randomly numbered and sorted. The evaluation was performed separately in a sensory evaluation room without communication among the evaluators. Different samples were evaluated at 5-min intervals, and the evaluators gargled with pure water in between each evaluation. The final score was the weighted average of the evaluation results from the 10 participants. A 9-point hedonic scale was used to determine the four attributes: 1 = dislike extremely, 2 = dislike very much, 3 = dislike moderately, 4 = dislike slightly, 5 = neither like nor dislike, 6 = like slightly, 7 = like moderately, 8 = like very much, and 9 = like extremely.

### 2.6. The Determination of Volatile Flavor Substances

The determination of volatile flavor substances was carried out according to the method of Zhou, Chong, Ding Gu et al. [15] with a slight modification. Briefly, broken sausages (5.0 g) were accurately weighed and transferred into a 20 mL vial (Supelco, Bellefonte, PA, USA). An amount of 5 μL of cyclohexanone (1 μg/mL) was added to each vial of sample as internal standard and then sealed with a polypropylene screw cap. The HS-SPME fiber (75 μm, carboxen/polydimethylsiloxane; Supelco, Bellefonte, PA, USA) was carefully inserted into the vial and kept 2~3 cm from the samples to avoid damage that could affect the analysis. The volatile compounds were extracted and collected with fibers at 50 °C for 30 min. The fiber was desorbed in the GC injector port for 3 min at 250 °C. The volatile flavor substances were identified using a DSQ II GC-MS mass spectrometer (Thermo Electron, Waltham, MA, USA). Chromatographic determination involved the following conditions: capillary column of 30 m length × 0.25 mm inner diameter × 0.25 mm film thickness (TR-5 MS, thermo Scientific, Waltham, MA, USA); helium flow rate of 1 mL/min; the temperature at 40 °C for 3 min, then at 3 °C/min to 70 °C, and 5 °C/min to 180 °C. Finally, the temperature was heated to 280 °C at 10 °C/min, and kept for 5 min. The conditions for mass spectrometry were as follows: ion source temperature, 230 °C; transfer line temperature, 250 °C; ionization mode, EI+; electron energy, 70 eV; scanning mass range 30–550 *m*/*z*. The mass spectrometry data of the flavor components were compared with the data from the NIST 08 database. Using cyclohexanone as the internal standard, a semi-quantitative method was used to identify the flavor components. The composition of the sample was quantified by comparing the peak areas of the flavor components with the internal standard peak area. 

### 2.7. The Determination of Total Volatile Base Nitrogen 

The total volatile base nitrogen was determined according to Dabadé et al. [16] with property modifications. The broken sausage samples (10 g) were added to 100 mL of distilled water to stir and dissolve them. After filtration using ordinary filter paper, 20 mL of the filtrate was placed in a Kjeldahl tube, 1 g of magnesium oxide was added, and then a K9860 automatic Kjeldahl apparatus (Haineng, Jinan, China) was used to determine the nitrogen content.

### 2.8. Oxidation Stability

#### 2.8.1. Protein Oxidation Measurement

Protein oxidation was evaluated by measuring the protein carbonyl and free sulfhydryl groups contents. The protein carbonyl content was measured according to the method described by Xu et al. [17] with a minor modification. To the broken sausage samples (2 g), we added 18 mL of normal saline that was homogenized twice with an IKA 25 high-speed homogenizer (Ultra Turrax, Staufen, Germany) at 12,000 rpm for 30 s each time, and centrifuged at 12,000 rpm for 5 min. The filtrates (2.0 mL) were taken from two groups; to one 0.5 mL of 2 mol/L hydrochloric acid solution was added to create a control, and to the other, 0.5 mL of 0.02 mol/L 2, 4-dinitrophenylhydrazine solution (dissolved in 2 mol/L hydrochloric acid) was added to create the determination samples. Thereafter, the mixture was reacted in a water bath at 37 °C for 15 min. The precipitate was washed thrice with 1 mL ethanol and ethyl acetate (*v*/*v* = 1/1). The precipitate was dissolved in 2 mL of 6 mol/L guanidine hydrochloride in a 37 °C water bath for 15 min. The absorbance was measured at 370 nm. The protein content was determined using the biuret method, with bovine serum protein as the standard. The carbonyl content was calculated using the following formula and expressed in nmol/mg protein.
$$\mathrm{Carbonylcontent}=\frac{\left(A_{370\mathrm{sample}}-A_{370\mathrm{control}}\right)\times 2000}{C_{\mathrm{protein}}\times 22.0}$$
where A_370sample_ indicates absorbance of sample at 370 nm; A_370control_ indicates absorbance of control at 370 nm; C_protein_ indicates protein concentration.

The content of the protein-free sulfhydryl group was determined according to the method described by Qing et al. [18] with slight modifications. To the smashed sausage samples (2 g), 18 mL of 0.1 mol/L tris-glycine buffer (pH 8.0) was added, and the mixture was homogenized twice at 12,000 rpm with an IKA 25 high-speed homogenizer (Ultra Turrax, Staufen, Germany) for 30 s each time. An aliquot of 0.5 mL of homogenate was added to 2.5 mL of 0.1 mol/L tris-glycine buffer (pH 8.0), and then added to 20 μL of 0.4% 5, -dithio-2-nitrobenzoic acid solution (0.1 mol·L^−1^ tris-glycine buffer, pH 8.0). For the blank control, 20 μL of 0.1 mol/L tris-glycine buffer was incubated in the dark at 37 °C for 15 min. After the reaction, the sample was centrifuged at 10,000 rpm for 5 min using a high-speed homogenizer. The absorbance of the supernatant was measured at a wavelength of 370 nm. The free sulfhydryl content of the samples was calculated using the following formula:$$\mathrm{Sulfhydrylcontent}\left(\mathrm{nmol}/\mathrm{mg}\;\mathrm{protein}\right)=75.53\times \left(A_{412\mathrm{sample}}-A_{412\mathrm{control}}\right)\times \frac{\mathrm{total}\;\mathrm{volume}}{\mathrm{sample}\;\mathrm{volume}}/C\mathrm{protein}$$

#### 2.8.2. The Determination of Lipid Oxidation

Lipid oxidation was measured according to the method described by Fu et al. [19] with suitable modifications. To 2 g of broken sausages, 18 mL of 0.9% normal saline was added, and the mixture was homogenized twice with an IKA 25 high-speed homogenizer (Ultra Turrax, Staufen, Germany) at 12,000 rpm for 30 s each time. To a 0.2 mL portion of homogenate, 0.2 mL 8.1% sodium dodecyl sulfate, 1.5 mL 20% acetate buffer, 1.5 mL 0.8% 2-thiobarbituric acid solution, and 0.6 mL distilled water was added. After a 60 min period for the reaction to occur at 90 °C, the water was fully cooled. A spectrophotometer was used to measure the absorbance at 532 nm. Various concentrations of 1, 1, 3, 3-malondialdehyde solution were used to create a standard curve, and the results were expressed as milligram of malondialdehyde per gram of sausage. 

### 2.9. Statistical Analysis

SAS 8.2 software was used for the statistical analysis of the data, and Origin 9.0 software was used for plotting. The mean ± standard deviation was used to represent the test data. ANOVA was used to observe the differences between different treatment groups, and Duncan’s multiple comparisons test was used to analyze the differences in the mean values of the samples (*p* < 0.05).

## 3. Results and Discussion

### 3.1. Effects of Cherry Powder Addition on Physicochemical Properties

The physical properties of the sausage are mainly in terms of the pH value, moisture content, color, and texture. The pH value, moisture content, color, and texture of the sausages after 0 day and 30 days of storage are shown in Table 1. As shown in Table 1, at day 0, the pH value of the sausages made with 3% and 5% cherry powder was lower than that of the control and 1% cherry powder groups (*p* ˂ 0.05). This indicates that the addition of cherry powder could reduce the pH value of sausage. After 30 days of storage, the pH value of the sausage was significantly lower than that of the control group (*p* < 0.05), regardless of the level of cherry powder supplementation (1%, 3%, or 5%). In line with our results, Jin, Choi, Jeong, and Kim [20] reported that the pH of pork sausage treated with cloves and ascorbic acid was significantly lower than that of the control during storage. This phenomenon might be attributed to the antioxidant property of cherry powder, cloves, and ascorbic acid, which inhibited the microbial or enzymatic degradation of the protein in the sausage, thus leading to the decrease in pH [21]. In addition, the decrease in pH might be caused by carbon monoxide reacting with the water within the patties to form carbon dioxide during storage [22].

The addition of cherry powder had a certain effect on the moisture content of the sausages, but there was no significant difference between the supplemental level of the cherry powder and the control group (*p* < 0.05). The moisture content of the sausage decreased significantly with the increase in the concentration of cherry powder (*p* < 0.05). This was so regardless of whether it was after 0 day or 30 days of storage. This indicated that the addition of too much cherry powder was not good for water retention within the sausages. This could be attributed to protein-water interactions, which allow water to seep into the myofibrillar proteins being placed by protein-polyphenol interactions, resulting in the loss of hydrogen-bonded water molecules, which could hinder the water-retaining ability of the meat proteins [23,24].

Color is the primary quality characteristic used by consumers to evaluate the quality of fresh meat, which directly affects their desire to purchase [25]. The addition of cherry powder significantly affected the color of the sausages. After sausage storage for 0 day, the *L**, *a**, and *b** values of the sausages were significantly increased with an increase in cherry powder addition when compared with those of the control group (*p* < 0.05). This may be due to the color of the cherry powder. When the sausage was stored for 30 days, with the increase in cherry powder supplementation level, the *b** value of the sausage was significantly increased, compared with that of the control group (*p* < 0.05), but there was no significant difference between the *L** and *a** values of the sausages (*p* < 0.05). This indicates that the addition of cherry powder has a good protective effect on the color of sausage during storage. Prommachart et al. (2020) [26] demonstrated that the incorporation of black rice water extracts retarded the reduction in the *L**, *a**, and *b** values of ground beef patties during chilled storage, due to the higher polyphenol and anthocyanin contents. Similarly, Turgut, Işıkçı, and Soyer (2017) [27] also reported that the rich anthocyanin and phenolic content of pomegranate extracts delayed the discoloration of pork patties during storage. Hence, the addition of a high concentration of antioxidants had a positive effect on improving the color stability of meat products. 

The addition of cherry powder also significantly affected the texture of the sausages. After storage for 0 day, the addition of cherry powder did not change the cohesion of the sausages. The hardness, elasticity, stickiness, and chewiness of the sausage were significantly decreased with the increase in cherry powder concentration, except for the fact that these properties in the 1% cherry powder had no significant difference when compared with those of the control group (*p* < 0.05). This indicates that the addition of cherry powder reduced the hardness of the sausage. Compared with the control group, after 30 days of storage, the hardness, elasticity, cohesion, adhesiveness, and chewiness of the 5% cherry powder group were significantly decreased (*p* < 0.05); nevertheless, there were no significant differences in the hardness, elasticity, cohesion, adhesiveness, and chewiness of the 1% and 3% cherry powder groups (*p* < 0.05). Our results agree with the results of Zhang Lin, Leng, Huang, and Zhou [28] who demonstrated that the addition of sage extract reduced the hardness of the sausage. Xu, Zhu, Liu, and Cheng [17] also reported that the incorporation of mulberry polyphenol reduced the hardness of dried minced pork slices during storage. In the control group, protein oxidation within the sausage may lead to the formation of cross-linking complexes and aggregation [29], whereas antioxidants might retard protein and lipid oxidation, thus maintaining the integrity of muscle fibers and reducing the hardness of the sausage after the cherry powder addition. Therefore, it is generally thought that the addition of antioxidant polyphenols protects the texture from damage caused by protein oxidation.

### 3.2. Effects of Cherry Powder Additive on Sensory Evaluation

As seen in Table 2, whether stored for 0 day or 30 days, when 1% cherry powder was added, the color, flavor, texture, and overall acceptability of the sausage were significantly higher than those of the control group (*p* ˂ 0.05). With subsequent increases in cherry powder concentration, the color, flavor, texture, and overall acceptability of the sausage were significantly lower than those of the control group (*p* ˂ 0.05). The results show that the addition of 1% cherry powder was beneficial to improving the sensory evaluation value of sausage, whereas the addition of a higher concentration of cherry powder negatively affected the color and taste of sausage due to the color and taste of the powder. In our study, the sensory evaluation and texture measurement results are consistent. However, our results are not similar to those reported by Zhang Lin, Leng, Huang, and Zhou [28] who indicated that, except for the color, the addition of sage during storage did not affect the flavor, texture, or overall acceptability of the sausage.

### 3.3. Effects of Cherry Powder Addition on Volatile Flavor Substances

The flavor of the sausages is mainly due to fat oxidation, flavoring, and protein degradation. As shown in Table 3, the characteristic volatile substances in Jiangsu-type sausage mainly include alcohols, aldehydes, ketones, acids, esters, and hydrocarbons. It can also be seen from Table 3 that the addition of cherry powder has a significant effect on the volatile flavor substances in sausage, and the overall flavor substances in sausage increase or decrease with the increase in storage time. When the samples were stored for 0 day or 30 days, the sausage contained 1-butanol, 1, 2-propanedio1, 2-methyl-hexadecanol, acetaldehyde, heptanal, 2-undecenal, nonanal, 2-octenal, nonanal, 2-decimal, hexanal, 3-methyl-butanal, acetic acid, oleic acid, hexacosene, heptane, hexadecane, and heptaethylene glycol monododecyl ether, whose contents were not significantly different from those of the control group (*p* < 0.05). Compared with the control group, the content of these substances decreased with the increase in concentration (*p* < 0.05). As the cherry powder concentration increased, the contents of 1-pantanol, hexanol, ethanol, benzaldehyde, pentanoic acid, caproic acid, palmitic acid, octane, hexadecane, 1, 3-hexadiene, octene, hexanoic acid ethyl ester, caproic acid vinyl ester, hexadecanoic acid methyl ester, and 2-naphthol in the sausages increased significantly when compared with those of the control subjects (*p* < 0.05). The decrease in some alcohols might be caused by esterification or oxidation reactions from processing the sausage, while the increase in other alcohols may be caused by lipid oxidation [30]. The decrease in some aldehydes might be caused by the oxidation of other substances, whereas the increase in other aldehydes may be caused by the oxidation and degradation of fats [31]. The increase or decrease in acids might be related to lipid oxidation and hydrolysis, while the change in hydrocarbons was closely related to lipid oxidation, degradation, and lipase activity [32]. In addition, the content of esters increased during the storage process and with the increase in the addition of cherry powder. This might be caused by the hydrolysis and oxidation of fat to produce fatty acids, and the esterification reaction of fatty acids and alcohols to produce esters [33]. Therefore, when the 1% cherry powder was added, the main aromatic components of the sausage were alcohols, aldehydes, and esters, and the overall flavor of the sausage significantly improved. 

### 3.4. Effects of Cherry Powder Addition on Total Volatile Base Nitrogen (TVB-N)

To a certain extent, the TVB-N reflects the degree of protein decomposition by endogenous proteases and microorganisms. The increase in TVB-N value is the result of the combined action of endogenous proteases and spoilage bacteria [16]. As shown in Figure 1, the initial TVB-N value in this study was similar to previous studies, i.e., approximately, 6 mg/100g [1]. When stored for 0, 10, 20, or 30 days, with an increase in cherry powder concentration, the TVB-N value in sausage decreased significantly (*p* ˂ 0.05). After 30 days of storage, 1% cherry powder reduced TVB-N by 26% compared to the control group. This may be because the polyphenols in cherry powder have a certain inhibitory effect on microbial growth and reproduction, and can inhibit the activity of the endogenous protease. Our results show a similar tendency to what was seen in the results of Liu, Tsau, Lin, Jan, and Tan [34] and Xiang, Cheng, Zhu, and Liu [1]. They indicated that after the addition of rosemary and mulberry polyphenols, the sausages had a lower TVB-N value, due to their antimicrobial activity, in comparison with the control. Compared to the previous report [1], lower TVB-N value during storage was observed in this study. These variations may be due to differences in pork varieties, processes, and storage conditions. 

### 3.5. Effects of Cherry Powder Addition on Protein Oxidation and Lipid Oxidation

Because cysteine residues in meat products are sensitive to oxidation, carbonyl and free sulfhydryl groups are usually used to evaluate protein oxidation [2]. The main reason for protein carbonylation in meat products is the oxidative modification of some side-chain amino acids, such as proline, arginine, lysine, and histidine residues, which are oxidized to form irreversible carbonyl components [35]. As shown in Figure 2, the protein carbonyl values of the 1% and 3% cherry powder on day 0 were not different from those of the control group (*p* > 0.05). Regardless of whether it were day 0, day 10, day 20, or day 30, an increase in cherry powder concentration brought about significant decreases in the protein carbonyl values of the sausages, compared with those of the control (*p* > 0.05). This could be due to the inhibition of protein oxidation by polyphenols in the cherry powder during storage. According to Xiang et al., phenolics are able to scavenge free radicals and chelate metal ions, thus exerting antioxidant function. Cheng, Liu, Zhang, Chen, and Wang [36] found that the carbonyl content of dried minced pork slices enriched with concentrated mulberry juice was lower than that of control samples during storage. Jia, Kong, Liu, Diao, and Xia [37] also suggested that black currant extract significantly inhibited the carbonyl formation of pork patties during storage. In addition to this, Andres, Petron, Adamez, Lopez, and Timon [38] also demonstrated good antioxidant properties for tomato and red grape extracts in lamb meat patties. Similarly, with the increase in cherry powder concentration, the value of the free sulfhydryl group increased significantly (*p* ˂ 0.05), except that the free sulfhydryl group value of 1% cherry powder was not different from that of the control group on day 0 (*p* > 0.05). This indicates that the addition of cherry powder effectively inhibited the protein oxidation of sausages. The reduction in the sulfhydryl content in the control samples was mainly due to the conversion of cysteine sulfhydryl groups to disulfide bonds [39]. Therefore, the polyphenols in cherry powder could scavenge free radicals efficiently and, subsequently, retard the loss of the cysteine sulfhydryl group [40].

Rancidity can produce unacceptable flavors, and negatively influence consumer acceptance of meat products [4]. In this study, lipid oxidation was evaluated using thiobarbituric acid values. The changes in thiobarbituric acid values directly reflect the degree of oxidative rancidity in meat products [41]. Figure 3 shows that the thiobarbituric acid value of sausage decreased significantly with an increase in cherry powder concentration (*p* ˂ 0.05). An exception to this is the fat oxidation value of 1% cherry powder on days 0, 10, 20, and 30, where there was no difference between the two groups (*p* > 0.05). These results indicate that the addition of cherry powder also effectively inhibited the lipid oxidation of sausages, and the effective inhibition of lipid oxidation might be due to the presence of cherry powder polyphenols. The polyphenols in cherry powder can effectively prevent lipid oxidation by providing hydrogen atoms to free radicals. İlyasoğlu [42] reported similar results, in which meatballs with different concentrations of rosehip powder (1%, 2%, and 4%) were cooked and stored for 9 days. The lipid oxidation of the cooked meatballs was effectively reduced by the dose of rose hip powder compared with that of the control group. Jia, Kong, Liu, Diao, and Xia [37] also reported similar results, which showed that polyphenols could inhibit lipid oxidation in pork patties by blocking radical chain reactions. Vattem, Randhir, and Shetty [43] indicated that cranberry powder reduced the malondialdehyde content in porcine muscle due to the influence of cranberry phenolics on the regulation of the cellular antioxidant enzyme response in oxidatively stressed porcine muscle. Our results are consistent with the above-mentioned findings and indicate the potential application of cherry powder as a natural antioxidant to protect meat products against lipid oxidation during storage. 

## 4. Conclusions

In conclusion, the results of the present study indicate that the addition of cherry powder effectively inhibited protein and lipid oxidation in sausages, slowed the increase in volatile base nitrogen during storage, and regulated the flavor and sensory quality of sausages. The proper dosage of cherry powder (1%) could improve the flavor and quality of sausages without affecting their physicochemical properties and sensory qualities. Excessive doses of cherry powder (3%) increased antioxidant capacity, and inhibited protein and lipid oxidation, resulting in a decrease in the overall sensory quality of the product. Therefore, cherry powder may be a promising antioxidant with potential applications in the sausage industry. In the future, the antioxidant mechanism of cherry powder in sausages during storage requires further study.

## Figures and Tables

**Figure 1 foods-11-03590-f001:**
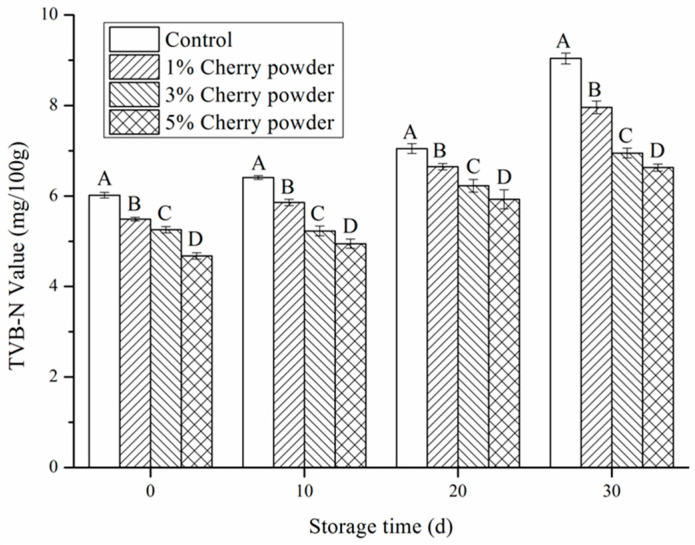
Effects of cherry powder addition on TVB-N values of Jiangsu-type sausage during storage. Note: Different letters (A, B, C, D) indicate a significant difference (*p* < 0.05).

**Figure 2 foods-11-03590-f002:**
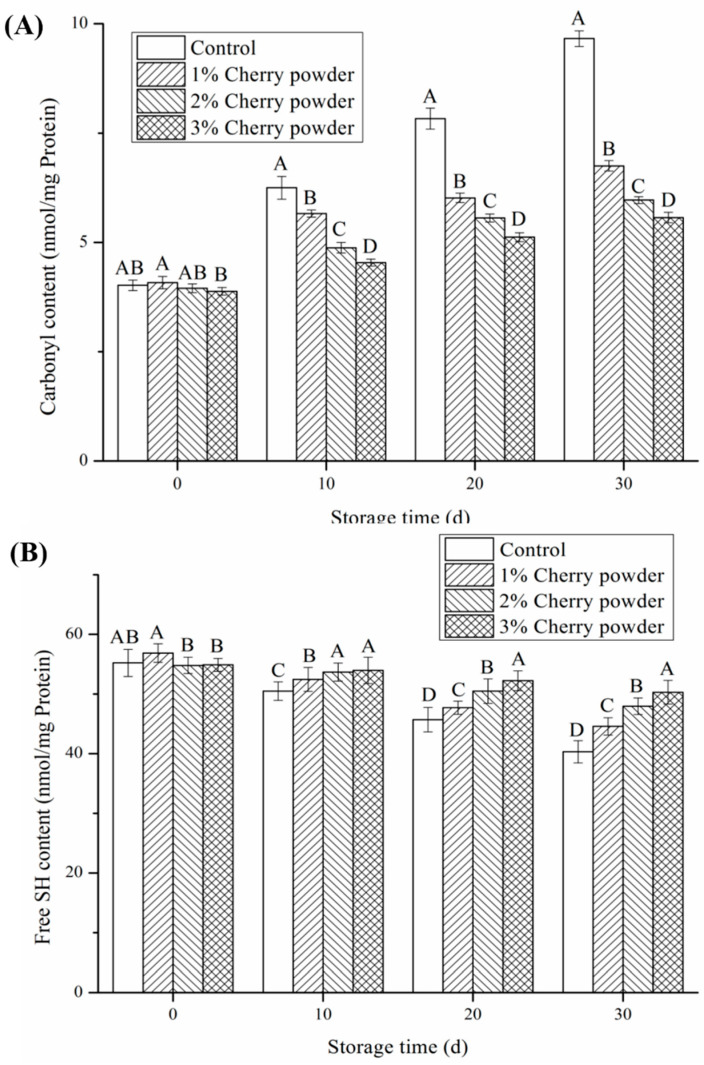
Effects of cherry powder addition on protein oxidation of Jiangsu-type sausage during storage. (**A**): carbonyl content, (**B**) Free SH content. Note: Different letters (A, B, C, D) indicate a significant difference (*p* < 0.05).

**Figure 3 foods-11-03590-f003:**
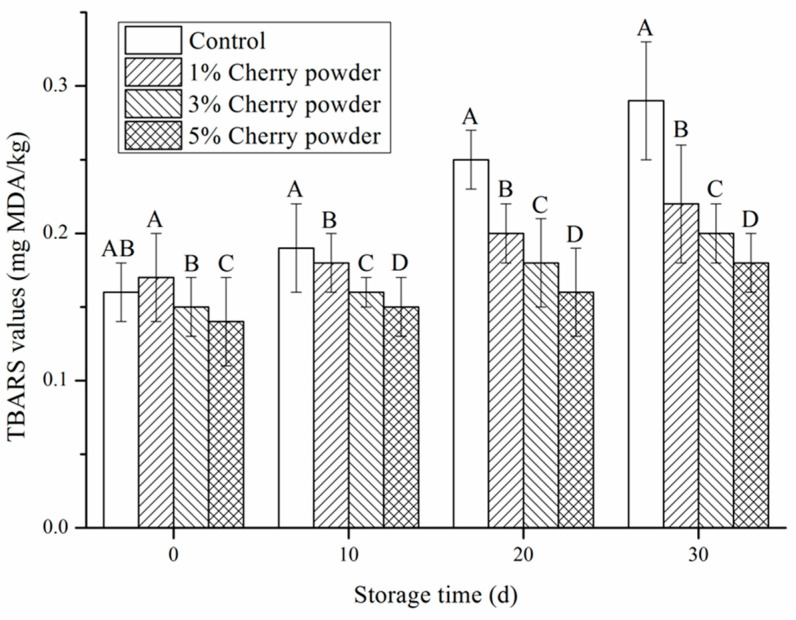
Effects of cherry powder addition on TBARS values of Jiangsu-type sausage during storage. Note: Different letters (A, B, C, D) indicate a significant difference (*p* < 0.05).

**Table 1 foods-11-03590-t001:** Effects of cherry powder addition on the texture properties of Jiangsu-type sausage during storage.

Indicator	Amount of Cherry Powder (%, 0 d)				Amount of Cherry Powder (%, 30 d)			
	0 (Control)	1%	3%	5%	0 (Control)	1%	3%	5%
pH	5.95 ± 0.01 ^C^	5.93 ± 0.02 ^C^	5.88 ± 0.01 ^D^	5.87 ± 0.01 ^D^	6.22 ± 0.01 ^A^	6.13 ± 0.02 ^B^	6.10 ± 0.01 ^B^	6.08 ± 0.01 ^B^
Moisture content/%	20.83 ± 0.27 ^B^	20.28 ± 0.13 ^B^	17.70 ± 0.14 ^E^	18.51 ± 0.30 ^D^	22.66 ± 0.67 ^A^	21.98 ± 0.82 ^A^	18.30 ± 0.49 ^D^	19.86 ± 0.39 ^C^
*L**	49.59 ± 0.22 ^B^	47.14 ± 0.18 ^C^	46.99 ± 0.45 ^D^	45.12 ± 0.62 ^E^	50.66 ± 0.12 ^A^	50.48 ± 0.45 ^A^	50.02 ± 0.56 ^A^	49.92 ± 0.45 ^A^
*a**	15.85 ± 0.63 ^C^	16.23 ± 0.30 ^D^	17.38 ± 0.28 ^A^	17.61 ± 0.27 ^A^	16.88 ± 0.29 ^AB^	17.55 ± 0.35 ^A^	16.76 ± 0.25 ^B^	17.26 ± 0.53 ^A^
*b**	13.39 ± 0.49 ^D^	14.26 ± 0.21 ^C^	14.83 ± 0.26 ^C^	14.77 ± 0.48 ^C^	16.21 ± 0.43 ^B^	16.90 ± 0.51 ^AB^	17.12 ± 0.71 ^A^	17.54 ± 0.30 ^A^
Hardness/N	7199 ± 142 ^C^	7312 ± 78 ^C^	6641 ± 88 ^D^	6089 ± 30 ^E^	8011 ± 104 ^A^	8013 ± 24 ^A^	8149 ± 121 ^A^	7749 ± 130 ^B^
Springiness/cm	0.83 ± 0.02 ^A^	0.82 ± 0.02 ^AB^	0.81 ± 0.01 ^B^	0.79 ± 0.02 ^C^	0.83 ± 0.02 ^A^	0.84 ± 0.05 ^A^	0.80 ± 0.02 ^B^	0.78 ± 0.02 ^C^
Cohesiveness	0.74 ± 0.02 ^A^	0.76 ± 0.01 ^A^	0.75 ± 0.01 ^A^	0.74 ± 0.01 ^A^	0.74 ± 0.02 ^A^	0.73 ± 0.01 ^A^	0.73 ± 0.01 ^A^	0.71 ± 0.01 ^B^
Gumminess/N	5426 ± 11 ^B^	5418 ± 0.57 ^B^	4968 ± 69 ^C^	4503 ± 45 ^D^	5939 ± 96 ^A^	5868 ± 46 ^A^	5963 ± 136 ^A^	5485 ± 102 ^B^
Chewiness/N × cm	4436 ± 41 ^B^	4350 ± 62 ^B^	4035 ± 17 ^C^	3541 ± 18 ^D^	5017 ± 32 ^A^	4949 ± 48 ^A^	4850 ± 98 ^B^	4296 ± 79 ^C^

Notes: Different letters A, B, C, D, and E in the same column indicate significant differences (*p* < 0.05).

**Table 2 foods-11-03590-t002:** Effects of cherry powder addition on sensory attributes of Jiangsu-type sausage during storage.

Indicator	Amount of Cherry Powder (%, 0 d)				Amount of Cherry Powder (%, 30 d)			
	0	1%	3%	5%	0	1%	3%	5%
Color	8.22 ± 0.18 ^A^	8.15 ± 0.45 ^A^	7.82 ± 0.62 ^B^	7.32 ± 0.55 ^D^	7.49 ± 0.31 ^C^	7.52 ± 0.18 ^C^	7.36 ± 0.52 ^D^	7.13 ± 0.33 ^E^
Flavor	8.17 ± 0.39 ^A^	8.20 ± 0.61 ^A^	8.25 ± 0.57 ^A^	7.92 ± 0.38 ^B^	7.99 ± 0.49 ^B^	8.02 ± 0.53 ^B^	7.96 ± 0.66 ^B^	7.83 ± 0.46 ^C^
Texture	8.04 ± 0.48 ^B^	8.30 ± 0.52 ^A^	8.02 ± 0.39 ^B^	7.82 ± 0.49 ^C^	7.89 ± 0.65 ^C^	8.02 ± 0.37 ^B^	7.86 ± 0.71 ^D^	7.68 ± 0.42 ^D^
Overall acceptability	7.72 ± 0.32 ^C^	8.26 ± 0.24 ^A^	7.61 ± 0.59 ^C^	6.67 ± 0.27 ^DE^	7.22 ± 0.21 ^C^	7.94 ± 0.14 ^B^	6.99 ± 0.13 ^D^	6.35 ± 0.39 ^E^

Notes: Different letters A, B, C, D, and E in the same column indicate significant differences (*p* < 0.05).

**Table 3 foods-11-03590-t003:** Effects of cherry powder addition on volatile flavor components of Jiangsu-type sausage during storage.

Compound Name	Cas#	Retention Time (min)	Amount of Cherry Powder (%, 0 d)				Amount of Cherry Powder (%, 30 d)			
			0	1%	3%	5%	0	1%	3%	5%
1-butanol	71-36-3	8.77	0.83 ± 0.05 ^A^	0.81 ± 0.10 ^A^	0.75 ± 0.12 ^B^	0.72 ± 0.07 ^B^	0.81 ± 0.04 ^A^	0.72 ± 0.16 ^B^	0.69 ± 0.03 ^C^	0.65 ± 0.17 ^C^
1-pantanol	71-41-0	13.01	2.41 ± 0.38 ^D^	2.54 ± 0.21 ^C^	2.56 ± 0.14 ^C^	2.60 ± 0.31 ^C^	2.34 ± 0.13 ^D^	2.37 ± 0.13 ^D^	2.79 ± 0.24 ^B^	2.98 ± 0.36 ^A^
Hexanol	111-27-3	17.31	2.01 ± 0.53 ^DE^	2.47 ± 0.22 ^CD^	2.93 ± 0.37 ^B^	3.53 ± 0.56 ^A^	1.83 ± 0.15 ^E^	2.33 ± 0.52 ^D^	2.64 ± 0.86 ^C^	2.93 ± 0.98 ^B^
Ethanol	64-17-5	3.19	7.56 ± 1.01 ^C^	7.81 ± 1.12 ^C^	8.07 ± 1.20 ^B^	8.16 ± 2.52 ^B^	8.10 ± 0.28 ^B^	8.17 ± 0.31 ^B^	8.29 ± 0.83 ^A^	8.49 ± 0.31 ^A^
1,2-propanediol	21994-81-0	5.82	0.14 ± 0.06 ^A^	0.13 ± 0.02 ^A^	0.11 ± 0.03 ^B^	0.10 ± 0.02 ^B^	0.11 ± 0.03 ^B^	0.09 ± 0.01 ^B^	0.07 ± 0.02 ^C^	0.05 ± 0.02 ^C^
2-methyl-hexadecanol	2490-48-4	9.37	0.41 ± 0.06 ^A^	0.36 ± 0.08 ^A^	0.22 ± 0.05 ^C^	0.21 ± 0.04 ^C^	0.31 ± 0.06 ^B^	0.28 ± 0.12 ^B^	0.21 ± 0.24 ^C^	0.23 ± 0.35 ^C^
Acetaldehyde	75-07-0	1.49	2.08 ± 0.15 ^A^	1.89 ± 0.32 ^A^	1.22 ± 0.58 ^C^	0.99 ± 0.05 ^D^	1.70 ± 0.12 ^B^	1.57 ± 0.56 ^B^	1.08 ± 0.57 ^C^	0.95 ± 0.10 ^D^
Heptanal	111-7-17	10.01	2.63 ± 0.14 ^A^	2.43 ± 0.26 ^B^	2.25 ± 0.57 ^C^	2.21 ± 0.63 ^C^	2.40 ± 0.10 ^B^	2.38 ± 0.18 ^B^	2.03 ± 0.21 ^D^	1.78 ± 0.14 ^E^
2-undecenal	2463-77-6	35.08	1.87 ± 0.19 ^A^	1.74 ± 0.18 ^B^	1.66 ± 0.21 ^C^	1.62 ± 0.32 ^C^	1.80 ± 0.23 ^A^	1.69 ± 0.23 ^B^	1.61 ± 0.14 ^C^	1.49 ± 0.32 ^D^
Nonanal	124-19-6	18.57	2.60 ± 0.43 ^A^	2.36 ± 1.11 ^AB^	2.45 ± 0.68 ^B^	2.26 ± 0.87 ^C^	2.46 ± 1.20 ^B^	2.41 ± 0.85 ^B^	2.16 ± 0.58 ^C^	2.09 ± 0.35 ^C^
2-octenal	2548-87-0	19.85	2.15 ± 1.02 ^A^	1.91 ± 0.06 ^B^	1.84 ± 0.25 ^C^	1.85 ± 0.47 ^C^	2.00 ± 0.14 ^AB^	1.93 ± 0.21 ^B^	1.94 ± 0.28 ^B^	1.95 ± 0.14 ^B^
3-methyl-butanal	590-86-3	2.81	0.30 ± 0.19 ^B^	0.26 ± 0.08 ^BC^	0.23 ± 0.20 ^D^	0.20 ± 0.09 ^D^	0.39 ± 0.04 ^A^	0.34 ± 0.06 ^B^	0.32 ± 0.07 ^B^	0.29 ± 0.36 ^B^
Benzaldehyde	100-52-7	23.45	1.42 ± 0.09 ^D^	1.47 ± 0.18 ^D^	1.62 ± 0.21 ^C^	1.73 ± 0.30 ^B^	1.56 ± 0.09 ^C^	1.80 ± 0.86 ^B^	2.09 ± 0.65 ^A^	2.15 ± 0.32 ^A^
2-decenal	3193-81-3	29.57	1.22 ± 0.24 ^C^	1.13 ± 0.53 ^C^	1.02 ± 0.68 ^D^	1.00 ± 0.12 ^D^	1.53 ± 0.24 ^A^	1.50 ± 0.31 ^A^	1.49 ± 0.27 ^A^	1.35 ± 0.20 ^B^
Hexanal	66-25-1	6.41	35.55 ± 1.62 ^A^	36.2 ± 2.28 ^A^	34.51 ± 1.09 ^AB^	33.02 ± 2.36 ^B^	29.33 ± 0.81 ^C^	29.61 ± 0.35 ^C^	28.05 ± 0.64 ^D^	28.73 ± 0.82 ^D^
Pentanal	110-62-3	3.82	3.27 ± 1.20 ^A^	3.23 ± 1.13 ^A^	3.12 ± 0.69 ^B^	2.98 ± 1.02 ^C^	3.15 ± 0.86 ^AB^	2.86 ± 0.54 ^CD^	2.71 ± 0.32 ^D^	2.56 ± 0.46 ^E^
Acetic acid	585-05-7	1.83	2.08 ± 0.56 ^A^	2.02 ± 0.47 ^A^	1.95 ± 0.68 ^B^	1.88 ± 0.32 ^B^	1.70 ± 0.08 ^C^	1.67 ± 0.12 ^C^	1.46 ± 0.25 ^D^	1.42 ± 0.36 ^D^
Oleic acid	22393-88-0	36.96	0.16 ± 0.02 ^A^	0.15 ± 0.09 ^A^	0.13 ± 0.06 ^B^	0.12 ± 0.03 ^B^	0.18 ± 0.05 ^A^	0.17 ± 0.06 ^A^	0.16 ± 0.03 ^A^	0.15 ± 0.08 ^A^
Erucic acid	112-86-7	46.74	0.05 ± 0.02 ^A^	0.04 ± 0.02 ^A^	0.03 ± 0.01 ^A^	0.03 ± 0.02 ^A^	0.04 ± 0.01 ^A^	0.03 ± 0.01 ^A^	0.03 ± 0.01 ^A^	0.05 ± 0.02 ^A^
Pentanoic acid	63169-61-9	30.33	0.94 ± 0.16 ^E^	1.32 ± 0.28 ^C^	1.52 ± 0.67 ^B^	1.57 ± 0.63 ^B^	1.18 ± 0.12 ^D^	1.34 ± 0.06 ^C^	2.25 ± 0.36 ^A^	2.18 ± 0.23 ^A^
Hexanoic acid	123-66-0	12.10	2.15 ± 0.12 ^C^	2.22 ± 0.33 ^C^	3.17 ± 1.69 ^B^	5.01 ± 1.58 ^A^	2.15 ± 0.10 ^C^	2.25 ± 0.74 ^C^	3.67 ± 1.41 ^B^	5.81 ± 0.58 ^A^
Hexadecenoic acid	2091-29-4	16.92	0.25 ± 0.09 ^A^	0.19 ± 0.10 ^B^	0.16 ± 0.05 ^C^	0.13 ± 0.08 ^C^	0.23 ± 0.07 ^A^	0.16 ± 0.02 ^B^	0.10 ± 0.03 ^D^	0.07 ± 0.04 ^D^
Palmitic acid	506-33-2	31.00	0.20 ± 0.06 ^C^	0.24 ± 0.09 ^AB^	0.28 ± 0.12 ^A^	0.26 ± 0.15 ^A^	0.23 ± 0.08 ^AB^	0.28 ± 0.06 ^A^	0.31 ± 0.16 ^A^	0.30 ± 0.13 ^A^
Octane	111-6-59	1.83	0.83 ± 0.16 ^C^	0.97 ± 0.07 ^A^	0.94 ± 0.12 ^A^	0.89 ± 0.24 ^AB^	0.87 ± 0.05 ^B^	0.91 ± 0.04 ^AB^	0.87 ± 0.08 ^B^	0.80 ± 0.09 ^C^
Hexadecane	7735-39-9	12.51	0.17 ± 0.08 ^D^	0.38 ± 0.14 ^BC^	0.44 ± 0.09 ^B^	0.52 ± 0.32 ^A^	0.22 ± 0.08	0.28 ± 0.07 ^C^	0.31 ± 0.03 ^BC^	0.35 ± 0.05 ^B^
Hexacosene	18835-33-1	20.99	0.14 ± 0.06 ^A^	0.08 ± 0.03 ^AB^	0.06 ± 0.02 ^B^	0.05 ± 0.02 ^B^	0.10 ± 0.04 ^A^	0.05 ± 0.01 ^B^	0.03 ± 0.01 ^C^	0.03 ± 0.01 ^C^
1, 3-hexadiene	74752-97-9	19.15	0.67 ± 0.18 ^D^	0.77 ± 0.02 ^C^	0.85 ± 0.12 ^B^	0.86 ± 0.20 ^B^	0.79 ± 0.09 ^C^	0.89 ± 0.26 ^AB^	0.94 ± 0.18 ^A^	0.92 ± 0.10 ^A^
Octene	6971-40-0	10.77	4.98 ± 0.85 ^B^	5.09 ± 0.86 ^AB^	5.24 ± 0.69 ^A^	5.17 ± 1.84 ^A^	4.93 ± 1.06 ^B^	4.90 ± 1.22 ^B^	5.13 ± 0.85 ^A^	5.31 ± 1.36 ^A^
Heptane	18835-33-1	20.99	0.60 ± 0.13 ^B^	0.57 ± 0.08 ^B^	0.51 ± 0.12 ^C^	0.53 ± 0.08 ^C^	0.70 ± 0.05 ^A^	0.67 ± 0.18 ^A^	0.57 ± 0.10 ^B^	0.59 ± 0.13 ^B^
Octodecane	55282-12-7	5.82	0.10 ± 0.03 ^AB^	0.12 ± 0.04 ^A^	0.05 ± 0.01 ^C^	0.09 ± 0.02 ^B^	0.11 ± 0.02 ^A^	0.14 ± 0.05 ^A^	0.04 ± 0.01 ^C^	0.06 ± 0.02 ^C^
Hexanoic acid ethyl ester	123-66-0	12.08	3.61 ± 0.82 ^D^	3.88 ± 1.25 ^D^	4.71 ± 0.52 ^C^	5.21 ± 0.98 ^B^	4.66 ± 1.02 ^C^	4.72 ± 0.65 ^C^	5.43 ± 0.41 ^B^	6.10 ± 0.13 ^A^
Caproic acid vinyl ester	3050-69-9	30.34	1.32 ± 0.19 ^C^	1.32 ± 0.10 ^C^	1.52 ± 0.16 ^B^	1.57 ± 0.21 ^B^	1.34 ± 0.08 ^C^	1.38 ± 0.22 ^C^	2.25 ± 0.84 ^A^	2.58 ± 0.51 ^A^
Octanoic acid, ethyl ester	106-32-1	20.45	0.32 ± 0.05 ^D^	0.33 ± 0.05 ^D^	0.32 ± 0.10 ^D^	0.38 ± 0.08 ^C^	0.46 ± 0.07 ^B^	0.47 ± 0.21 ^B^	0.45 ± 0.08 ^B^	0.57 ± 0.12 ^A^
Octadecadiynoic acid, methyl ester	18202-24-9	32.38	0.33 ± 0.07 ^D^	0.61 ± 0.07 ^B^	0.64 ± 0.19 ^B^	0.70 ± 0.12 ^A^	0.23 ± 0.04 ^E^	0.53 ± 0.06 ^C^	0.53 ± 0.11 ^C^	0.60 ± 0.08 ^B^
2-naphthol	91253-94-0	21.45	0.10 ± 0.02 ^C^	0.12 ± 0.05 ^C^	0.14 ± 0.05 ^B^	0.16 ± 0.05 ^B^	0.12 ± 0.03 ^C^	0.15 ± 0.02 ^B^	0.18 ± 0.06 ^A^	0.20 ± 0.03 ^A^
Heptaethylene glycol monododecyl ether	3055-97-8	59.89	0.14 ± 0.02 ^A^	0.14 ± 0.06 ^A^	0.11 ± 0.05 ^A^	0.09 ± 0.02 ^B^	0.12 ± 0.02 ^AB^	0.10 ± 0.03 ^B^	0.09 ± 0.02 ^B^	0.08 ± 0.03 ^B^

Notes: Different letters A, B, C, D, and E in the same column indicate significant differences (*p* < 0.05).

## Data Availability

Data is contained within the article.

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
