# Peer review of "Effects of Cherry (Prunus cerasus L.) Powder Addition on the Physicochemical Properties and Oxidation Stability of Jiangsu-Type Sausage during Refrigerated Storage"

_foods, 2022, doi:10.3390/foods11223590_

Round 1
Reviewer 1 Report
The main objective is to determine the percentage of cherry powder to improve the sensory and physico-chemical properties in the case of Jiangsu-type sausage
The subject can be considered relevant, following the antioxidant effect of cherry powder from a different perspective compared to research published so far.
Determination of volatile flavor substances and total volatile base nitrogen, together with protein oxidation and lipid oxidation measurement provides a substantial amount of new information, well-discussed, which I have not found in other previous studies.
I believe that a characterization of the cherry powder would be necessary.
I believe that the authors must specify:
from which fruit was cherry powder obtained? Prunus cerasus? Cherry powder can be confused with acerola cherry powder (Malpighia emarginata), because it is often called cherry powder and is important due to its vitamin C content.
the antioxidant properties of the cherry powder used, because they influence the percentage of powder that is added to obtain the results presented
Apart from the previously mentioned, I don't think that any other analyses are necessary, because the results are sufficient to recommend the addition of 1% of cherry powder.
Are the values obtained for TVB-N and its variation over the 30 days, in the case of the control sample, similar to those obtained by other authors?
The conclusions are consistent with the research objectives and the results obtained. I think from this section it is clear why 1% cherry powder is recommended, even if some results are better at 3% cherry powder
Especially if, in the "introduction" section, the suggested information is entered, I consider the references to be adequate
· page 1, line 3: correct ”refregerated” in ”refrigerated”
· page 7, line 243: correct ”Soviet sausage”
Author Response
Response to the reviewer’s comments
Reviewer #1:
The main objective is to determine the percentage of cherry powder to improve the sensory and physico-chemical properties in the case of Jiangsu-type sausage. The subject can be considered relevant, following the antioxidant effect of cherry powder from a different perspective compared to research published so far. Determination of volatile flavor substances and total volatile base nitrogen, together with protein oxidation and lipid oxidation measurement provides a substantial amount of new information, well-discussed, which I have not found in other previous studies.
Response: Thanks for your comments.
- Comment: I believe that a characterization of the cherry powder would be necessary. I believe that the authors must specify: from which fruit was cherry powder obtained? Prunus cerasus? Cherry powder can be confused with acerola cherry powder (Malpighia emarginata), because it is often called cherry powder and is important due to its vitamin C content.
Response: Thanks for your comments and helpful suggestion. The cherry powder is characterised by good antioxidant activity due to it is rich in polyphenols such as hydroxycinnamates, flavonols, flavan-3-ols, and especially anthocyanins. To better show this characterization, we have added literatures that identified the bioactive components of cherry (Lines 41-43 in the revised manuscript). The cherry powder used in this study was obtained from Prunus cerasus L. Have added this information in the revised version to make it clear (Lines 1, 38 in the revised manuscript).
2 Comment: The antioxidant properties of the cherry powder used, because they influence the percentage of powder that is added to obtain the results presented. Apart from the previously mentioned, I don't think that any other analyses are necessary, because the results are sufficient to recommend the addition of 1% of cherry powder. Are the values obtained for TVB-N and its variation over the 30 days, in the case of the control sample, similar to those obtained by other authors? The conclusions are consistent with the research objectives and the results obtained. I think from this section it is clear why 1% cherry powder is recommended, even if some results are better at 3% cherry powder. Especially if, in the "introduction" section, the suggested information is entered, I consider the references to be adequate.
Response: Thanks for your helpful suggestions. Have added the antioxidant properties of cherry powder and references (Line 41-43 in the revised manuscript). The initial TVB-N value in this study was similar to previous studies, i.e., approximately, 6 mg/100g. [1] Compared to the control group, 1% cherry powder inhibit 26% of TVB-N after 30 days of storage. Compared to other authors [1], lower TVB-N value during storage was observed in this study. These variations may be due to differences in pork varieties, processes, and storage conditions.
3 Comment: line 3: correct "refregerated" in "refrigerated" 3
Response: Corrected. (Line 3 in the revised manuscript)
- 4. Comment: line 243: correct Soviet sausage
Response: Corrected. (Line 250 in the revised manuscript)
References
- Xiang, R.; Cheng, J.R., Zhu, M.J., & Liu, X.M. Effect of mulberry (Morus alba) polyphenols as antioxidant on physiochemical properties, oxidation, and bio-safety in Cantonese sausages. LWT - Food Sci Technol.2019, 116, doi: 108504. 10.1016/j.lwt.2019.108504.

Reviewer 2 Report
The work concerns the influence of cherry powder on the physicochemical properties of sausage. Natural additives, especially those that do not affect the taste of the food, and additionally allow to protect the nutritional value, seem to be worth the attention of producers and consumers.
Below is a list of comments that I propose to take into account when preparing a revised version of the manuscript.
Introduction
Please provide relevant literature to support the statements quoted in the introduction: Line 41/42 (Some studies….quercetin [?]); Line 42/43 (Therefore….antiapoptosis [?]);
The sentence in Line 49-51 is not clear, please complete it or redraw “The results of their experiment showed that all anthocyanins had an obvious scavenging effect (???), the correlation coefficients were greater than those of the antioxidant, and the anthocyanins were also more effective (???).
Material and methods
Line 66 and 66: Please indicate which mentioned reagents were analytically pure and which were “biochemical reagents”? Can the purity of the biochemical reagents be determined more precisely?
Line 119 – 121: Please provide more details on how the headspace extraction was performed, as well as how the volatile flavor substances were collected.
Line 123 - What GC column was used (name, producer)?
Line 125 - What was the volume of the injection?
Line 126-127 The transfer line temperature is missing.
Line 128 - What was the retention time of cyclohexanone used as IS, how much and when it was introduced into the samples. It was not previously listed among the reagents (Line 60-65).
Line 150 and 160 - please standardize the way formulas are given. Please also explain what the values inserted in the formulas mean.
Line 165 - What does SDS mean?
Line 165 - please check if the record is correct: “1.5 mL 20% pH 3.5 acetate buffer” (? 20% pH?)
Line 160 – “One mole per liter of 1, 1, 3, 3-malondialdehyde solution was used as the standard curve” ? I guess this solution was used to create the standard curve?
Results and discussion
Line 183-186 Missing in this passage is a brief explanation of why cloves and ascorbic acid behave similarly to cherry powder
Table 3 on page 9 lists “hexanoic acid” with retention time 12.10 min and on page 10 its ester with 12.08 min. Were these compounds really able to be separated on the “5ms” GC column? Please check if there is a mistake? Probably the peak at time 12.08-12.10 comes from only one of these compounds. Esters are less polar and should not have the same retention time as the corresponding acids.
Unfortunately, basing only on the NIST database may not be accurate. Merely confirming the result through analysis of standards can solve such a problem.
Line 280/281 – please add reference
Other minor comments
Page 2, line 45 – should be “Nowak et al”
Page 3, line 86 – please insert: … “and pH was” measured using…
Page 4, line 138 – the word …sulfhydryl ”groups” is missing in the sentence.
Line 248 “3-methyl-butanalmyl” - please check if there is any mistake here. I guess it should be “3-methyl-butanal” ?
Author Response
Reviewer #2:
The work concerns the influence of cherry powder on the physicochemical properties of sausage. Natural additives, especially those that do not affect the taste of the food, and additionally allow to protect the nutritional value, seem to be worth the attention of producers and consumers. Below is a list of comments that I propose to take into account when preparing a revised version of the manuscript.
Response: Thanks for your comments.
- Comment: Introduction
Please provide relevant literature to support the statements quoted in the introduction: Line 41/42 (Some studies…quercetin [?]); Line 42/43 (Therefore .antiapoptosis [?]);
Response: Have added as suggestion. (Lines 41, 44 in the revised manuscript)
- Comment:The sentence in Line 49-51 is not clear, please complete it or redraw “The results of their experiment showed that all anthocyanins had an obvious scavenging effect (???), the correlation coefficients were greater than those of the antioxidant, and the anthocyanins were also more effective (???).
Response: Thanks for your suggestion. Have removed this unclearly expressed in the revised manuscript.
- Comment:Material and methods
Line 66 and 66: Please indicate which mentioned reagents were analytically pure and which were “biochemical reagents”? Can the purity of the biochemical reagents be determined more precisely?
Response: Have indicated in the revised version. (Lines 60-63 in the revised manuscript)
- Comment:Line 119 – 121: Please provide more details on how the headspace extraction was performed, as well as how the volatile flavor substances were collected.
Response: Have provided as suggestion. (Lines 117-123 in the revised manuscript)
- Comment:Line 123 - What GC column was used (name, producer)?
Response: Have provided the information, that is, TR-5 MS, Thermo Scientific, Waltham, Mass., U.S.A. (Line 125 in the revised manuscript)
- Comment:Line 125 - What was the volume of the injection?
Response: The volume of the injection was 5 μL. Have revised (Line 128 in the revised manuscript).
- Comment:Line 126-127 The transfer line temperature is missing.
Response: Transfer line temperature was 250 °C. Have revised (Line 129 in the revised manuscript).
- 8. Line 128 - What was the retention time of cyclohexanone used as IS, how much and when it was introduced into the samples. It was not previously listed among the reagents (Line 60-65).
Response: Retention time=17.70; the amount of cyclohexanone is 5 μL, and it was introduced into the sample before headspace extraction. Have revised (Lines 117-123 in the revised manuscript).
- 9. Comment: Line 150 and 160 - please standardize the way formulas are given. Please also explain what the values inserted in the formulas mean.
Response: Have revised. (Lines 153-156 in the revised manuscript)
- 10. Comment: Line 165 - What does SDS mean?
Response: It means sodium dodecyl sulfate. Have revised (Line 171 in the revised manuscript).
- 11. Comment: Line 165 - please check if the record is correct: “1.5 mL 20% pH 3.5 acetate buffer” (? 20% pH?)
Response: Thanks for your kind reminder. Have revised as 1.5 mL 20% acetate buffer (Line 171 in the revised manuscript).
- 12. Comment: Line 160 – “One mole per liter of 1, 1, 3, 3-malondialdehyde solution was used as the standard curve” ? I guess this solution was used to create the standard curve?
Response: Yes, it was used to create the standard curve. Have revised (Line 174 in the revised manuscript).
- 13. Comment:Results and discussion
Line 183-186 Missing in this passage is a brief explanation of why cloves and ascorbic acid behave similarly to cherry powder.
Response: We Have revised, that is, this phenomenon might be attributed to the antioxidant property of cherry powder, cloves, and ascorbic acid, which inhibited the microbial or enzymatic degradation of the protein in the sausage, thus leading to the decrease in pH. (Lines 191-193 in the revised manuscript).
- Comment:Table 3 on page 9 lists “hexanoic acid” with retention time 12.10 min and on page 10 its ester with 12.08 min. Were these compounds really able to be separated on the “5ms” GC column? Please check if there is a mistake? Probably the peak at time 12.08-12.10 comes from only one of these compounds. Esters are less polar and should not have the same retention time as the corresponding acids.
Unfortunately, basing only on the NIST database may not be accurate. Merely confirming the result through analysis of standards can solve such a problem.
Response: We apologize for the mistakes. The retention time of hexanoic acid ethyl ester was 12.08. We have revised.
- 15. Comment:Line 280/281 – please add reference
Response: Have revised. (Line 290 in the revised manuscript).
- 16. Comment:Other minor comments
Page 2, line 45 – should be “Nowak et al”
Response: Corrected. (Line 46 in the revised manuscript).
- 17. Comment:Page 3, line 86 – please insert: … “and pH was” measured using…
Response: Have revised as suggestion (Line 83 in the revised manuscript).
- 18. Comment:Page 4, line 138 – the word …sulfhydryl ”groups” is missing in the sentence.
Response: Corrected (Line 142 in the revised manuscript).
- 19. Comment:Line 248 “3-methyl-butanalmyl” - please check if there is any mistake here. I guess it should be “3-methyl-butanal”?
Response: Yes, it is 3-methyl-butanal. Corrected. (Line 255 in the revised manuscript)

Reviewer 3 Report
This research work presents a proposal of relevant knowledge for the sausage industry. However, I would like to make the following observations.
The title should be specific, what do you mean by physicochemical properties? It would also be important to separate in the section on materials and methods what corresponds to physicochemical and what corresponds to oxidation.
I also recommend placing the scientific name of the cherry.
Please check the spaces between sentences and after the period.
In line 17 the term addition is repetitive, check the whole summary.
In line 40 the word delicious is subjective, I recommend using another term.
Lines 55 to 57 seem compromising for the trials and degrees of substitution performed, for that sentence a complete and robust experimental design would be needed. I recommend rewording the objective.
Revise spacing between headings and subheadings.
Line 179, after storage for 0 days what does it refer to? there was no storage.
In Table 1 the word value is unnecessary, it is understood.
In Table 2 the term Jiangsu-type is not written in italics, revise the whole text.
Table 3 should be presented in a better way, also take care that the parentheses appear on the same line.
I recommend the authors to expand their discussion of the results.
Author Response
Reviewer #3:
This research work presents a proposal of relevant knowledge for the sausage industry. However, I would like to make the following observations. The title should be specific, what do you mean by physicochemical properties? It would also be important to separate in the section on materials and methods what corresponds to physicochemical and what corresponds to oxidation. I also recommend placing the scientific name of the cherry.
Response: Thanks for your comments and constructive suggestions. The physicochemical properties in this study are mainly involved in pH, water content, color, total volatile base nitrogen, texture, and flavor. Considering the length of the title, this study choose to use physicochemical properties instead of specific descriptions. In order to make the presentation of physicochemical properties more specific, we have revised the description in the method section (Line 79, 140, 141, 167 in the revised manuscript). We have revised the scientific name of the cherry, namely, cherry (Prunus cerasus L.) in the title and introduction (Line 1, 38 in the revised manuscript).
- 1. Comment: Please check the spaces between sentences and after the period.
Response: Thanks to your kind reminder, we have carefully checked the manuscript and made corrections.
- Comment: In line 17 the term addition is repetitive, check the whole summary.
Response: Have removed the repetitive “addition’’.
- 3. Comment: In line 40 the word delicious is subjective, I recommend using another term.
Response: Have revised as sweet. (Line 39 in the revised manuscript).
- 4. Comment: Lines 55 to 57 seem compromising for the trials and degrees of substitution performed, for that sentence a complete and robust experimental design would be needed. I recommend rewording the objective.
Response: Have rewritten the sentences to highlight the objective of this study. (Line 50-54 in the revised manuscript).
- 5. Comment: Revise spacing between headings and subheadings.
Response: Have revised as suggestion.
- 6. Comment: Line 179, after storage for 0 days what does it refer to? there was no storage.
Response: Have revised as at the 0 day. (Line 185 in the revised manuscript).
- 7. Comment: In Table 1 the word value is unnecessary, it is understood.
Response: Deleted as suggested. (Table 1 in the revised manuscript).
- 8. Comment: In Table 2 the term Jiangsu-type is not written in italics, revise the whole text.
Response: Have revised as suggestion. (Line 246, 250, 271, 310, 330 in the revised manuscript).
- Comment: Table 3 should be presented in a better way, also take care that the parentheses appear on the same line.
Response: Thanks to your suggestions. Have adjusted the font size, the distance between rows, the thickness of lines to make it better. (Table 3 in the revised manuscript).
- 10. Comment: I recommend the authors to expand their discussion of the results.
Response: Thanks for your helpful suggestions. Have expanded the discussion of the results to improve the quality of manuscript. (Lines 190-193, 276-285, 297-298 in the revised manuscript).

Round 2
Reviewer 2 Report
The article has been revised as indicated. The only comment concerns point 6.
According to the revised description, GC-MS analysis was performed using SPME fiber (this information was missing from the first version of the manuscript), so the information that the GC volume of injection was 5 ul (line 129) is unsubstantiated and can be removed.
Author Response
- Comment: According to the revised description, GC-MS analysis was performed using SPME fiber (this information was missing from the first version of the manuscript), so the information that the GC volume of injection was 5 ul (line 129) is unsubstantiated and can be removed.
Response: Thanks for your comments and deleted.
